# Selective Elimination of NG2-Expressing Hair Follicle Stem Cells Exacerbates the Sensitization Phase of Contact Dermatitis in a Transgenic Rat Model

**DOI:** 10.3390/ijms21186922

**Published:** 2020-09-21

**Authors:** Yasuhisa Tamura, Kumi Takata, Asami Eguchi, Yosky Kataoka

**Affiliations:** 1Laboratory for Cellular Function Imaging, RIKEN Center for Biosystems Dynamics Research, 6-7-3 Minatojima-minamimachi, Chuo-ku, Kobe 650-0047, Japan; kumi@riken.jp (K.T.); asami.eguchi@riken.jp (A.E.); kataokay@riken.jp (Y.K.); 2Multi-Modal Microstructure Analysis Unit, RIKEN-JEOL Collaboration Center, 6-7-3 Minatojima-minamimachi, Chuo-ku, Kobe 650-0047, Japan

**Keywords:** NG2, hair follicle stem cells, allergic contact dermatitis, anagen, hapten, immunosuppression

## Abstract

The hair cycle consists of three different phases: anagen (growth), catagen (regression), and telogen (resting). During the anagen phase, hair follicle stem cells (HFSCs) in the bulge and the secondary hair germ proliferate and generate the outer and inner root sheath cells and the hair shafts. We previously identified NG2-immunoreactive (NG2+) cells as HFSCs in both regions of the hair follicles. Recently, the interaction between the hair cycle and the cutaneous immune system has been re-examined under physiological and pathological conditions. However, the roles of NG2+ HFSCs in the skin’s immune system remain completely elucidated. In the present study, we investigated whether the elimination of NG2+ HFSCs affects the induction of allergic contact dermatitis, using a herpes simplex virus thymidine kinase (HSVtk)/ganciclovir (GCV) suicide gene system. When the GCV solution was applied to the skin of NG2-HSVtk transgenic (Tg) rats during the depilation-induced anagen phase, NG2+ HFSCs in the Tg rat skin induced apoptotic cell death. Under exposure of a hapten, the selective ablation of NG2+ HFSCs during the anagen phase aggravated the sensitization phase of allergic contact dermatitis. These findings suggest that NG2+ HFSCs and their progeny have immunosuppressive abilities during the anagen phase.

## 1. Introduction

The hair cycle in mammals consists of three distinct stages: anagen (growing phase), catagen (regressing phase), and telogen (resting phase) [1,2]. At the anagen phase, hair follicle stem cells (HFSCs) in the bulge region and secondary hair germ (sHG) cells proliferate and produce their progenies, including outer root sheath cells, inner root sheath cells, and hair shafts [3,4]. HFSCs in the bulge are composed of heterogeneous cell populations, including CD34^+^Lgr5^–^ cells and CD34^+^Lgr5^+^ cells in the upper and lower portions of the bulge, respectively [5,6]. By contrast, the sHG cells are immunopositive for Lgr5, but not CD34. We recently reported that all cell populations in the bulge and sHG are immunoreactive for NG2, also known as chondroitin sulfate proteoglycan 4 (CSPG4), and that the NG2-immunopositive (NG2+) cells in both regions of the hair follicles have proliferative ability [7].

There has been ongoing research interest focused on the interaction between hair follicle cycling and the skin immune system, which has been examined under both physiological and pathological conditions. The activation of HFSCs is known to be regulated by macrophages [8,9,10] and regulatory T-cells [11], whereas hair follicle-derived cytokines regulate the generation and maintenance of skin-resident memory T-cells [12]. In addition, the immunosuppressive effects of contact dermatitis during the anagen phase were demonstrated to be much higher than those during the telogen phase [13,14]. However, the specific roles of HFSCs during the anagen phase in the induction of contact dermatitis remains incompletely understood. In the present study, we investigated the effects of the selective ablation of NG2-expressing HFSCs on dinitrofluorobenzene (DNFB)-induced contact dermatitis in a rat model, using the herpes simplex virus thymidine kinase (HSVtk)/ganciclovir (GCV) system [15]. DNFB is a typical hapten that is widely used for establishing animal models of allergic contact dermatitis [16].

## 2. Results

### 2.1. Expression of HSVtk in NG2+ HFSCs of Tg Rats

In line with our previous study [7], we observed NG2+ cells in the bulge, sHG, and dermal papilla regions of the hair follicles during the telogen phase, and in the outer root sheath during the anagen and catagen phases in NG2-HSVtk Tg rats (data not shown). Moreover, similar to our previous study of the brain [15], the expression of the transgene HSVtk was only observed in the NG2+ cells of the hair follicles in the Tg rats (Figure 1). These results suggested that this Tg rat model was appropriate for eliminating the dividing NG2+ cells in the hair follicles.

### 2.2. GCV Induced Programmed Cell Death in the Dividing NG2+ Cells of the Hair Follicles during Anagen

HFSCs in the sHG and the bulge begin to proliferate upon entry to the anagen phase [3,4]. One day after the first application of GCV to the skin of Tg rats, almost all terminal deoxynucleotidyl transferase dUDP nick-end labeling (TUNEL)-positive cells (as a marker of apoptosis) were NG2+ cells in the sHG of the hair follicles (Figure 2A). Indeed, proliferation of HFSCs in the sHG was reported to precede the activation of stem cells in the bulge region of hair follicles at the beginning of the anagen phase [17]. In addition, the expression of caspase-3, as another marker of apoptosis, was observed in the NG2+ cells of the bulge regions at day 4 (Figure 2B). These results confirmed that GCV could selectively eliminate NG2+ cells in this model.

Moreover, we examined the morphology of the hair follicles following application of GCV or vehicle during the depilation-induced anagen phase. Many of the hair follicles in the vehicle-treated skin showed a middle anagen-like morphology at seven days after depilation (Figure 3A), whereas the hair follicles in the GCV-applied skin seemed to be preserved during the telogen stage or early anagen (Figure 3B). We confirmed that application of GCV solution did not affect the hair growth cycle of the hair follicles in wild type (WT) rats. In addition to histological studies, we investigated the gene expression of NG2 in both vehicle- and GCV-treated skin with a quantitative polymerase chain reaction assay. The expression of NG2 in GCV-treated skin was significantly lower than that in vehicle-treated skin (Figure 3C). These findings indicated that topical application of GCV to the skin of NG2-HSVtk Tg rats could trigger cell apoptosis in the dividing NG2-expressing HFSCs during the anagen phase.

### 2.3. Selective Elimination of NG2+ HFSCs during Anagen-Induced Contact Dermatitis with a Depilatory Cream

We used a hair clipper and depilatory cream to remove the hair shafts on the back skin of the rats and to induce anagen re-entry of the hair follicles. GCV or vehicle solution was added to the same depilated regions once a day five times in total (Figure 4A). The GCV-treated skin began to display weak redness within 2–3 days, showed severe cutaneous erythema at 3–4 days, and then the symptoms of contact dermatitis worsened at seven days after depilation (Figure 4B, Table 1; *n* = 15). By contrast, the vehicle-treated skin showed none of these phenomena for seven days under the same conditions (Figure 4C, Table 1; *n* = 10). Furthermore, both vehicle- and GCV-treated skin of WT and normal Wistar rats never showed these immune responses with treatment of depilatory cream (data not shown).

To verify whether the ablation of NG2+ HFSCs during the anagen phase was associated with the induction of immune responses, the hair shafts of Tg rats were shaved with hair clippers and then GCV was applied to the skin without further depilatory cream treatment (Figure 5A). In almost all cases, none of the biological responses observed with the depilatory cream were observed in the GCV-applied skin (Figure 5B, Table 1; *n* = 8). In addition, the vehicle-applied (control) skin without treatment of depilatory cream did not show any immune reactions (Figure 5C, Table 1; *n* = 5). These findings indicated that eliminating NG2+ HFSCs during the anagen phase alone did not induce immune reactions. Therefore, the symptoms of contact dermatitis, including skin erythema and dry, scaly skin, were specifically induced by the selective ablation of NG2+ HFSCs during anagen, under the exposure of hair depilatory cream.

### 2.4. Selective Ablation of NG2+ HFSCs during Anagen Exacerbated the Sensitization Phase of DNFB-Induced Contact Dermatitis

We next ablated NG2+ HFSCs during the anagen phase in Tg rats with DNFB-induced contact dermatitis. As shown in Figure 6A, the Tg rats were sensitized by topical application of 0.5% DNFB solution to a defined area of the back skin following depilation with electric clippers, with the subsequent application of GCV or vehicle solution to the same area. During DNFB sensitization, the GCV-applied skin began to display erythema on day one, which became more severe on day three, and then the symptoms of contact dermatitis were aggravated, including dry, cracked, and scaly skin on day seven (Figure 6B, Table 2; *n* = 4). By contrast, the vehicle-treated skin began to show redness on day one, exhibited similar responses up to day seven, and then the redness started to disappear on day nine (Figure 6C, Table 2; *n* = 4). These results demonstrated that the selective ablation of NG2-expressing HFSCs during the anagen phase provoked the exacerbation of contact dermatitis following exposure to DNFB, whereas only the topical application of 0.5% DNFB elicited redness, but never induced more severe symptoms, such as dry, cracked, and scaly skin, in the presence of NG2+ HFSCs and their progeny.

## 3. Discussion

Recently, there has been renewed focus on the interaction between HFSCs and the skin immune system [10,11]. In this study, we showed that the selective elimination of NG2+ HFSCs during depilation-induced anagen provoked the aggravation of contact dermatitis under exposure of DNFB or depilatory cream. DNFB is a well-known hapten and is also used to induce contact hypersensitivity in the establishment of an experimental rodent model of allergic contact dermatitis [18,19,20]. Contact hypersensitivity and allergic contact dermatitis comprise an initial sensitization phase, occurring at the first contact of the skin with the hapten, and a subsequent elicitation phase, which is induced by repeated contact with the same hapten in sensitized individuals [21,22,23]. To investigate the effects of the ablation of NG2-expressing HFSCs on the sensitization phase of contact dermatitis, Tg rats were treated with a single application of DNFB solution to the back skin, and the skin of the non-ablation group showed redness, which supports a previous report with normal mice [24]. However, the skin of the NG2+ HFSCs ablation group displayed more severe symptoms, including cutaneous erythema, and dry, cracked, scaly skin. It has been known that CD4+ or CD8+ T cells are associated with contact dermatitis induced by DNFB. In our model, we confirmed that the expression of CD4 and CD8 genes in GCV-treated skin was approximately 20- and 800-fold higher than that in vehicle-treated skin, respectively (Figure 7). These more severe symptoms were also observed in the NG2+ HFSCs ablation group under exposure to depilatory cream. We further confirmed that the selective elimination of NG2+ HFSCs alone, without DNFB or depilatory cream, did not induce the symptoms of contact dermatitis. Collectively, these results indicate that the selective elimination of NG2+ HFSCs during the anagen phase exacerbated the sensitization phase of contact dermatitis specifically.

With our study design, we could not completely rule out the possibility that the death-induced inflammatory responses of NG2+ HFSCs were associated with the aggravation of contact dermatitis. However, we found that the cell death of NG2+ HFSCs, induced by the HSVtk/GCV system, was apoptosis, based on the immunoreactivity of apoptotic markers, including activated caspase-3 and oligonucleosomal DNA fragmentation with the TUNEL assay. Moreover, we confirmed that the protein expression of inflammatory cytokines showed similar levels in the Tg rat skin of both GCV-treated and vehicle-applied groups under the treatment of a depilatory cream (*n* = 3). Specifically, the IL-6 concentrations in the GCV-applied skin at three and seven days were 205 ± 48 and 112 ± 16 pg/mg protein, whereas those in the vehicle-treated skin were 224 ± 26 and 164 ± 68 pg/mg protein, respectively. In addition, the expression level of IL-1a in the GCV-treated skin was lower than that in the vehicle-applied skin at three days (1.7 ± 0.7 vs. 4.3 ± 2.2 ng/mg protein) and seven days (0.9 ± 0.4 vs. 2.5 ± 0.2 ng/mg protein). These data suggested that the apoptotic cell death of NG2+ HFSCs likely does not promote inflammatory responses under our experimental conditions.

Overall, our findings suggest that NG2+ HFSCs and their progeny during anagen might have immunosuppressive effects against allergic sensitization of contact dermatitis; however, the detailed mechanism remains unclear. In previous reports, immunosuppressive activities in contact dermatitis during anagen were relatively high compared with those observed during the telogen phase [11,12]. Immunosuppressants, such as transforming growth factor-beta and the alpha-melanocyte-stimulating hormone, are produced from the hair follicles during the anagen phase [25,26,27,28,29,30]. Therefore, these molecules might play crucial roles in suppressing the exacerbation of contact dermatitis. However, further studies are needed to clarify the mechanisms of the immunosuppressive effects of HFSCs and their progeny.

## 4. Materials and Methods

### 4.1. Animals

NG2-HSVtk Tg rats, with the expression of the HSVtk transgene in NG2+ cells [15], were established from Wistar rats and obtained from Japan SLC (Shizuoka, Japan). The selective expression of the transgene was confirmed by a polymerase chain reaction-based genotyping with specific primers, as described previously [15]. All rats were housed in isolator cages with free access to food and water and were maintained under 12-h light/dark cycles under controlled temperature (20 ± 2 °C) and humidity (50 ± 10%). All experimental protocols were approved by the Ethical Committee on Animal Care and Use of the RIKEN Center for Biosystems Dynamics Research (MA2009-17-20) (1 April 2018), and were performed in accordance with the Principles of Laboratory Animal Care (NIH publication No. 85-23, revised 1985).

### 4.2. Selective Elimination of NG2+ HFSCs during Depilation-Induced Anagen

Hair follicles in adult rodents are synchronized during the first postnatal hair cycle, and after 6 weeks, hair cycling becomes gradually unsynchronized with age [31]. However, hair follicles in the telogen phase can be synchronously induced for anagen entry by the depilation of hair shafts on the back skin [31,32]. We used electric clippers to remove the hair shafts from the dorsal skin for inducing anagen entry, and also used a depilatory cream to penetrate the GCV solution into the skin (Figure 4A). After depilation, the rats were topically applied GCV or vehicle (saline) solution to the defined area of the depilated skin once a day for 4 days.

### 4.3. DNFB-Induced Contact Dermatitis

DNFB was diluted in an acetone:olive oil solution (4:1 vol/vol) immediately prior to use. Before application of DNFB, the hair shafts from the Tg rat skin were removed with electric clippers to induce the transition of the hair follicles from the telogen to the anagen phase. After depilation, 200 μL of 0.5% DNFB solution was applied to a defined rectangle area (20 × 25 mm) of the back skin of the rats, followed by treatment of 200 μL GCV or vehicle solution with a micropipette to the same area at least 30 min after DNFB application (Figure 6A).

### 4.4. Immunohistochemistry

The rats were deeply anesthetized using isoflurane and perfused transcardially with 4% formaldehyde buffered in 0.01 M phosphate-buffered saline (PBS; pH 7.4). The skin tissues were removed and post-fixed in a 4% formaldehyde solution in 0.01 M PBS at 4 °C overnight, then soaked in 30% sucrose solution. Skin sections (40-μm thickness) were prepared using a microtome and collected as free-floating sections. The skin sections were immunostained with polyclonal goat anti-HSVtk antibody (1:100; Santa Cruz Biotechnology, Santa Cruz, CA, USA) and monoclonal mouse anti-NG2 antibody (1:200; Millipore, Billerica, MA, USA), or polyclonal rabbit anti-NG2 antibody (1:200; Millipore), to confirm the expression of the HSVtk transgene in the NG2+ cells. In addition, to detect programmed cell death in NG2+ cells, skin sections were immunostained with polyclonal rabbit anti-activated caspase-3 antibody (1:500; Cell Signaling Technology, Beverly, MA, USA), or were stained using the Apoptag Plus Fluorescein In Situ Apoptosis Detection Kit (S7111; Sigma-Aldrich, Saint Louis, MO, USA). The skin sections were incubated with these primary antibodies at 4 °C for 12–18 h. After washing for 30 min with PBS, containing 0.3% Triton-X100, the skin sections were incubated with Cy2- or Cy3-conjugated secondary antibodies (1:200; Jackson ImmunoResearch, West Grove, PA, USA) at 4 °C for 3–4 h. The stained sections were mounted with Hoechst dye 33,258 (Nacalai Tesque Inc., Kyoto, Japan) and examined using a confocal laser-scanning microscope (EZ- c1; Nikon, Tokyo Japan).

### 4.5. Real-Time Polymerase Chain Reaction Assay

Total RNA was extracted from skin tissues of each Tg rat with an ISOGEN kit (NIPPON GENE, Tokyo, Japan). Using the PrimerScript RT reagent Kit with gDNA Eraser (Takara Bio, Shiga, Japan), RNA (1 μg) samples were reverse-transcribed into cDNA. cDNA samples from each skin tissue were amplified with the KAPA SYBR FAST Universal qPCR Kit (Kapa Biosystems, Wilmington, MA, USA) on a Thermal Cycle Dice Real Time system TP800 (Takara Bio). PCR primers used for the detection of NG2 and ribosomal protein S18 (rps18) were as follows: forward (5′-AGGTAAGCATGATGTCCAGGTG-3′), reverse (5′-CAGTTGTGAGTGGAATGGCTTG-3′) for NG2 and forward (5′-CTTCCACAGGAGGCCTACAC-3′), reverse (5′-GATGGTGATCACACGCTCCA-3′) for rps18. The rps18 was used as the reference gene for the normalization of gene expression levels.

### 4.6. Statistical Analysis

Data are presented as mean ±SEM. Statistical analysis of the data was performed by an unpaired *t*-test.

## Figures and Tables

**Figure 1 ijms-21-06922-f001:**
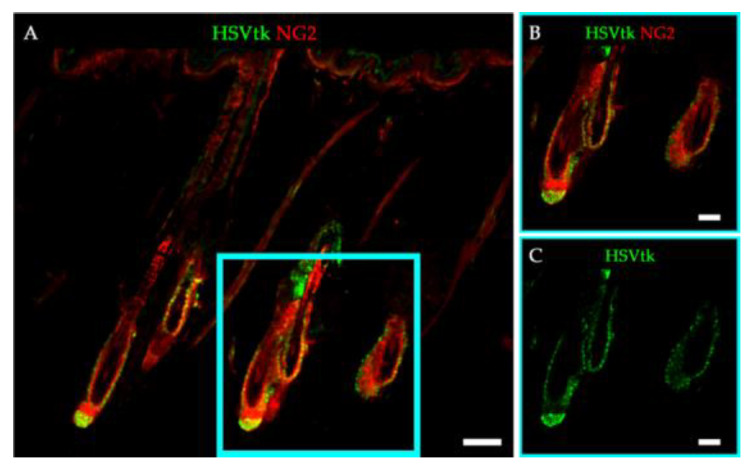
Expression of the HSVtk transgene in NG2+ cells in the hair follicles of NG2-HSVtk Tg rats. (**A**,**B**) Skin sections of NG2-HSVtk Tg rats stained with HSVtk (green) and NG2 (red) antibodies. (**B**) Magnified images of the area in the light blue square in (**A**). (**C**) Magnified images of immunofluorescence staining for HSVtk (green), shown as the light blue square area in (**A**). Scale bars, 100 μm (**A**), 50 μm (**B**,**C**).

**Figure 2 ijms-21-06922-f002:**
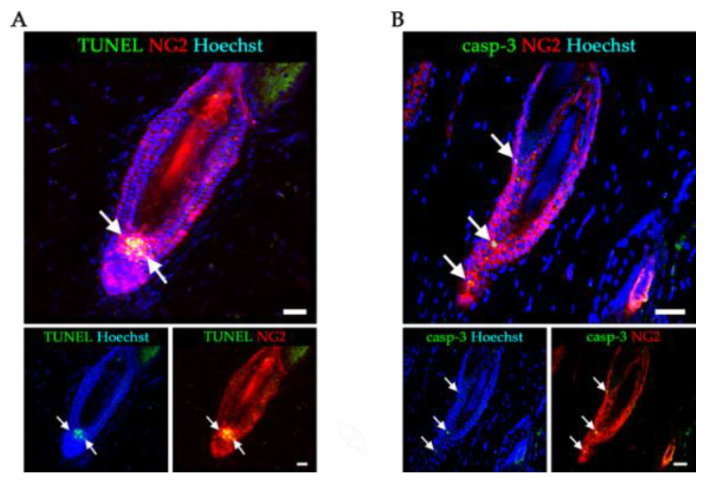
TUNEL and activated caspase 3 staining in NG2+ HFSCs following application of GCV to the skin of Tg rats. (**A**) Skin sections stained with TUNEL (green) and NG2 (red) antibodies. (**B**) Skin sections stained with activated caspase-3 (casp-3) (green) and NG2 (red) antibodies. Arrows in (**A**) and (**B**) indicate HFSCs co-expressing NG2 and markers of apoptotic cell death. Scale bars, 50 μm.

**Figure 3 ijms-21-06922-f003:**
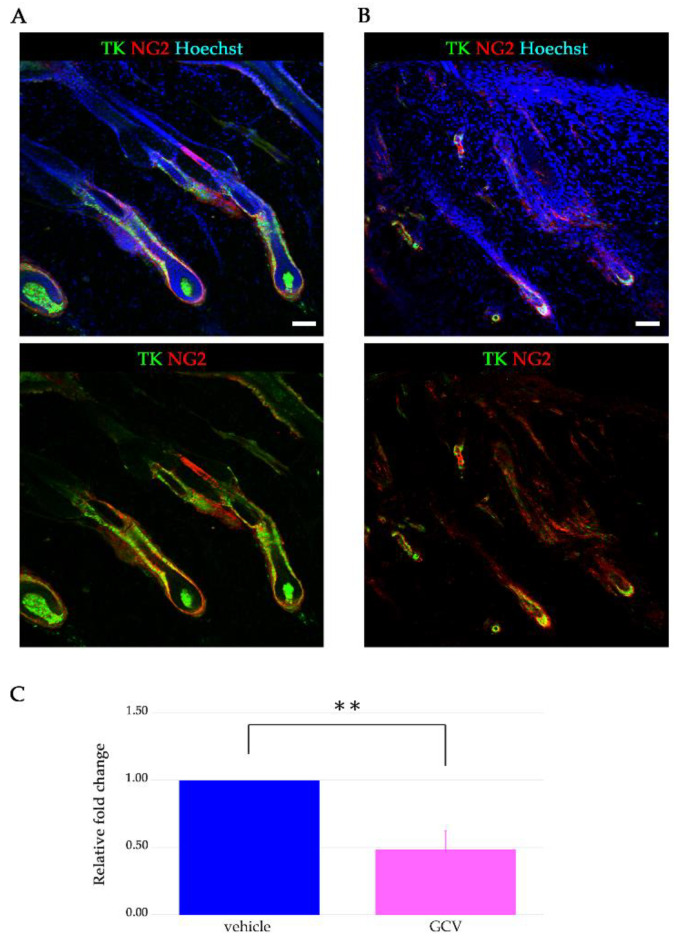
Morphology of the hair follicles in the GCV- or vehicle-applied skin of NG2-HSVtk Tg rats. (**A**,**B**) Immunostaining for HSVtk (green), NG2 (red), and Hoechst (blue) of vehicle-applied (**A**) and GCV-treated (**B**) skin sections. Upper panels: staining images for HSVtk (green), NG2 (red), and Hoechst (blue). Lower panels: staining images for HSVtk (green) and NG2 (red). Scale bars, 100 μm. (**C**) Gene expression of NG2 in vehicle-treated skin (blue bar, *n* = 4) and GCV-applied skin (pink bar, *n* = 4) of Tg rats at 7 days. Data are mean ± SEM. ** *p* < 0.01.

**Figure 4 ijms-21-06922-f004:**
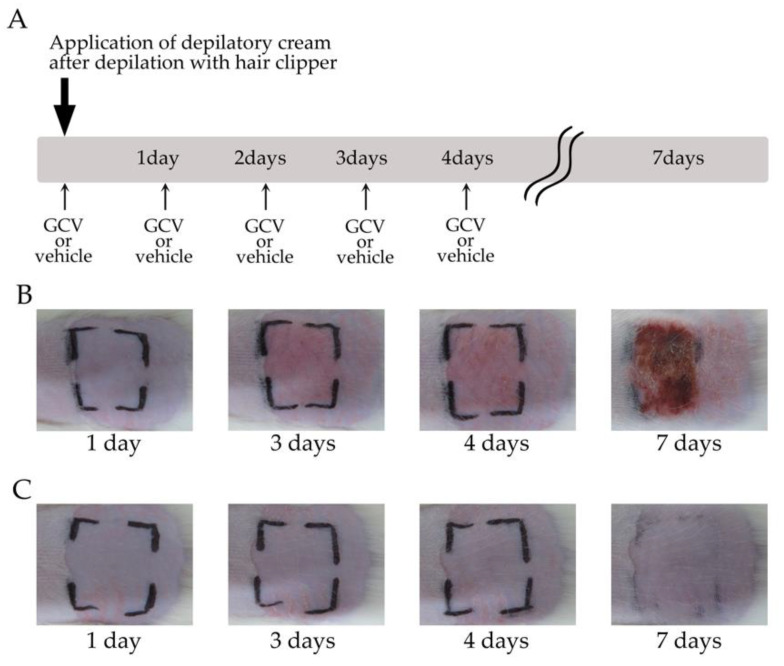
Selective ablation of NG2+ HFSCs during the depilation-induced anagen phase aggravated contact dermatitis with treatment of a depilatory cream. (**A**) Schematic of the experimental protocol. The bold arrow indicates depilation with a hair clipper and application of the depilatory cream. Arrows show application of GCV or vehicle to defined area (black lines) of the Tg rat skin. GCV, ganciclovir. (**B**,**C**) Photographs of GCV-applied (**B**) and vehicle-treated (**C**) skin of Tg rats at 1, 3, 4, and 7 days after the first application following treatment of the depilatory cream.

**Figure 5 ijms-21-06922-f005:**
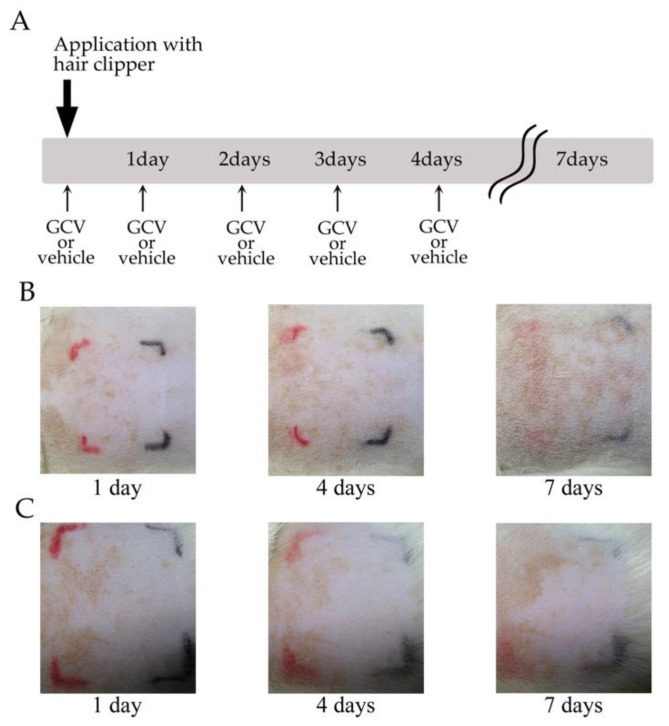
Selective ablation of NG2+ HFSCs during depilation-induced anagen did not induce symptoms of allergic contact dermatitis without application of depilatory cream. (**A**) Schematic of the experimental protocol. The bold arrow indicates depilation with a hair clipper. Arrows show application of GCV or vehicle to defined area (black lines) of the Tg rat skin. GCV, ganciclovir. (**B**,**C**) Photographs of GCV-applied (**B**) and vehicle-treated (**C**) skin of Tg rats at 1, 4, and 7 days.

**Figure 6 ijms-21-06922-f006:**
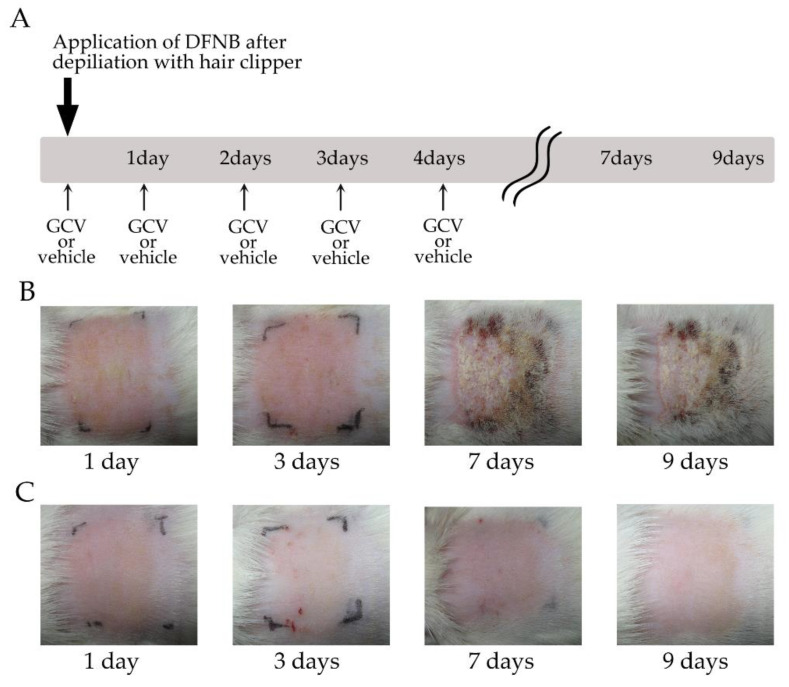
Selective elimination of NG2+ HFSCs during the depilation-induced anagen stage exacerbated the sensitization phase of contact dermatitis under exposure of DNFB. (**A**) Schematic of the experimental protocol. The bold arrow indicates depilation with hair clippers and application of 0.5% DNFB solution. Arrows show application of GCV or vehicle to defined area (black lines) of the Tg rat skin. GCV, ganciclovir. (**B**,**C**) Photographs of GCV-applied (**B**) and vehicle-treated (**C**) skin of Tg rats at 1, 3, 7, and 9 days after treatment of DNFB.

**Figure 7 ijms-21-06922-f007:**
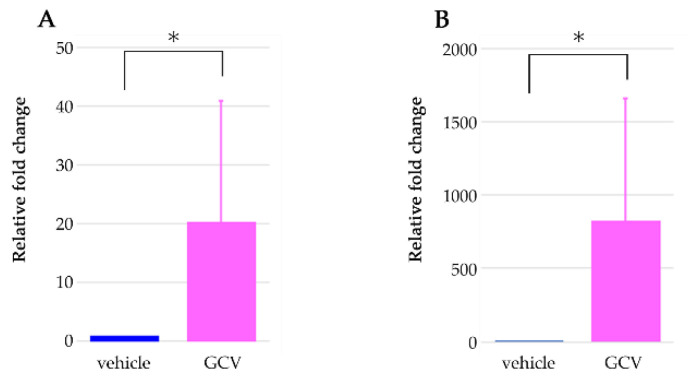
Expression levels of T-cell related genes. Gene expression of CD4 (**A**) and CD8 (**B**) in vehicle-treated skin (blue bars, n = 4) and GCV-applied skin (pink bars, *n* = 4) of Tg rats at 7 days. Data are mean ± SEM. * *p* < 0.05.

**Table 1 ijms-21-06922-t001:** Contact dermatitis is induced by selective ablation of NG2+ HFSCs with depilatory cream.

	Depilatory Cream	Redness (Erythema)	More Severe (Dry, Cracked, Scaly, Skin)
GCV group	+	15/15	13/15
Vehi group	+	0/10	0/10
GCV group	-	1/8	0/8
Vehi group	-	0/5	0/5

GCV group: GCV-treated rats. Vehi group: Vehicle-treated rats. The data represent the number of rats showing each symptom/the total number of animals.

**Table 2 ijms-21-06922-t002:** Effects of NG2+ HFSCs cell ablation on DNFB-induced contact dermatitis.

	DNFB	Redness (Erythema)	More Severe (Dry, Cracked, Scaly, Skin)
GCV group	+	4/4	4/4
Vehi group	+	4/4	0/4

GCV group: GCV-treated rats. Vehi group: Vehicle-treated rats. The data represent the number of rats showing each symptom/the total number of animals.

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
