# Peer review of "Selective Elimination of NG2-Expressing Hair Follicle Stem Cells Exacerbates the Sensitization Phase of Contact Dermatitis in a Transgenic Rat Model"

_ijms, 2020, doi:10.3390/ijms21186922_

Round 1

Reviewer 1 Report

The authors have previously identified NG2 as a marker of a population of hair follicle stem cells.

They now investigate whether elimination of NG2  affects the induction of allergic contact dermatitis using a herpes simplex virus thymidine kinase (HSVtk)/ganciclovir (GCV) suicide gene system.

1.The authors  examined the morphology  of the hair  follicles  following  application  of  GCV  or vehicle during the depilation-induced anagen phase.

They state that 'Many of the hair follicles in the vehicle-treated skin showed a middle anagen-like morphology at 7 days after depilation (Figure3A), whereas the hair follicles in the GCV-applied skin seemed to be preserved during the telogen stage or early anagen (Figure3B). These findings indicated that topical application of GCV to the skin of NG2-HSVtk Tg rats could trigger cell apoptosis in the dividing NG2-expressing HFSCs during anagen'

The control is vehicle alone with no GVC. Have the authors investigated the effects of GVC on the hair cycle of a non transgenic animal to rule out that it is not a toxic effect of GVC

2. Of more concern is the fact the authors report that  GCV-treated  deplitated (plucked) skin over vehicle treated skin   began  to  display  weak  redness within 2–3  days,  showed  severe cutaneous erythema  at  3–4  days,  and  then the  symptoms  of  contact  dermatitis worsened  at  7  days. have they carried out an appropriate control using a non transgenic rat. Depliation by plucking causes significant damage to the skin and can sometimes even result in death of the animal. GVC has numerous side effects including rash and skin irritation. this could be exasperated by the plucking as GVC could easily penetrate the skin causing irritation.

3. Likewise the authors report

that elimination of NG2+  HFSCs  during  anagen alone did  not induce immune reactions. Therefore, the symptoms of contact dermatitis, including skin erythema and dry, scaly skin, were specifically induced by the selective ablation of NG2+HFSCs during anagen under exposure of hair depilatory cream.

Hair depilatory creams are notorious for causing skin irritation. Therefore if they have plucked the skin to induce anagen then applied GVC and a depilatory cream there is a very high chance the cream will get into the epidermis and dermis causing significant irritation. have the authors used an appropriate control of a non transgenic animal with anagen induction via plucking and then exposure to GVC and depilatory cream.

This is essential to confirm this is specific to NG2 ablation and not just a non specific affect due to skin damage caused by plucking and subsequent exposure to GVC and depliatory cream

Author Response

Response to Reviewer 1 Comments

Thank you for your comments and suggestions.

We would like to reply to the comments as below.

We have revised our manuscript in all parts indicated in red in the revised version.

Point 1: The authors examined the morphology of the hair follicles following application of GCV or vehicle during the depilation-induced anagen phase. They state that 'Many of the hair follicles in the vehicle-treated skin showed a middle anagen-like morphology at 7 days after depilation (Figure3A), whereas the hair follicles in the GCV-applied skin seemed to be preserved during the telogen stage or early anagen (Figure3B). These findings indicated that topical application of GCV to the skin of NG2-HSVtk Tg rats could trigger cell apoptosis in the dividing NG2-expressing HFSCs during anagen.

The control is vehicle alone with no GVC. Have the authors investigated the effects of GVC on the hair cycle of a non-transgenic animal to rule out that it is not a toxic effect of GVC. 

Response 1: Thank you for your comments. We confirmed that application of GCV did not affect the hair growth cycle of hair follicles in wild type (WT) rats. We added this in the “Results” section (Line 88-89).

Point 2: Of more concern is the fact the authors report that GCV-treated deplitated (plucked) skin over vehicle treated skin began to display weak redness within 2–3 days, showed severe cutaneous erythema at 3–4 days, and then the symptoms of contact dermatitis worsened at 7 days. Have they carried out an appropriate control using a non transgenic rat. Depliation by plucking causes significant damage to the skin and can sometimes even result in death of the animal. GVC has numerous side effects including rash and skin irritation. this could be exasperated by the plucking as GVC could easily penetrate the skin causing irritation.

Response 2: Thank you for your comments. We confirmed that GCV-treated rat skin did not show immune responses, including rash and skin irritation in both WT and normal Wistar rats. We added this in the “Results” section (Line 107-109).

Point 3: Likewise, the authors report that elimination of NG2+ HFSCs during anagen alone did not induce immune reactions. Therefore, the symptoms of contact dermatitis, including skin erythema and dry, scaly skin, were specifically induced by the selective ablation of NG2+HFSCs during anagen under exposure of hair depilatory cream.

Hair depilatory creams are notorious for causing skin irritation. Therefore, if they have plucked the skin to induce anagen then applied GVC and a depilatory cream there is a very high chance the cream will get into the epidermis and dermis causing significant irritation. have the authors used an appropriate control of a non-transgenic animal with anagen induction via plucking and then exposure to GVC and depilatory cream.

This is essential to confirm this is specific to NG2 ablation and not just a non-specific affect due to skin damage caused by plucking and subsequent exposure to GVC and depilatory cream.

Response 3: Thank you for your suggestion. We confirmed that application of GCV and depilatory cream to skin following anagen re-entry did not show adverse responses in both wild type (non-transgenic) and normal Wistar rats (Line 107-109).

Reviewer 2 Report

In this manuscript, the author employed HSVtk/GCV suicide gene system to eliminate the NG2 positive HFSCs. Further study showed ablation of NG2+ HFSCs during anagen exacerbated the sensitization phase of DNFB-induced contact dermatitis. This work demonstrated the novel role of HFSCs in skin immune disease.  This finding is novel and interesting. A few issues should be addressed especially lacking a crucial mice control group.

  1. Although I believe the HSVtk/GCV suicide gene system, in Figure 3, it seems like there is no big difference of NG2 staining in two groups. Statistic analysis is needed.
  2. In figure 4, the authors showed application of GCV induced contact dermatitis in mice skin, however how can the authors exclude the individual influence of GCV? As far as I know, the side effects of application of GCV cream on human skin include redness, pain, itching, et al...
  3. In figure 5, the hair shafts of Tg rats were shaved with hair clippers and without further depilatory cream treatment, how could the mice skin be so clean even similar as Figure 4. Although the authors explained the depilatory cream is to penetrate the GCV solution into the skin, the detailed demonstration about this or other function of depilatory cream in this MS is needed in the results.
  4. Without depilatory cream treatment, the GCV did not induced dermatitis (Figure 5), however the GCV also did not enter the skin. Thus, a WT mice group with depilatory cream and GCV treatment is crucial.
  5. Figure 6, contact dermatitis marker gene expression analysis in two groups is helpful for the conclusion.

Author Response

Response to Reviewer 2 Comments

Thank you for your comments and suggestions.

We would like to reply to the comments as below.

We have revised our manuscript in all parts indicated in red in the revised version.

Point 1: Although I believe the HSVtk/GCV suicide gene system, in Figure 3, it seems like there is no big difference of NG2 staining in two groups. Statistic analysis is needed. 

Response 1: Thank you for the suggestion. According to your comments, we performed qPCR studies to assess the mRNA expression of NG2 in two groups: vehicle- and GCV-treated skin with depilatory cream. We showed that expression of NG2 in hair follicles of GCV treated skin was less than half of that in vehicle treated skin. Also, we described this in “Results” and “Methods” sections (Line 89-92 and Line 259-271).

Point 2: In figure 4, the authors showed application of GCV induced contact dermatitis in mice skin, however how can the authors exclude the individual influence of GCV? As far as I know, the side effects of application of GCV cream on human skin include redness, pain, itching, et al...

Response 2: Thank you for the suggestion. In our study, GCV application did not show inflammation and contact dermatitis in almost all. As far as I know, in clinical field, GCV is used as intravenous or ophthalmic medications for treatment of herpes virus infection, and also there is no GCV cream as external medicine until now. As you indicated, application of GCV ophthalmic gel and acyclovir ointment is used as antiviral medications and is reported to show some adverse events; punctate keratitis, conjunctival hyperemia and contact dermatitis (Contact Dermatitis. 2001, 44, 265-269; Ther Clin Risk Manag., 2014, 10, 665-681). However, several studies suggested that these adverse events were relatively rare case (Contact Dermatitis. 2001, 44, 265-269) and that the adverse events by acyclovir cream were caused by their ointment base (Bourezane Y et al., Allergy 1996, 51, 755–759).

Point 3: In figure 5, the hair shafts of Tg rats were shaved with hair clippers and without further depilatory cream treatment, how could the mice skin be so clean even similar as Figure 4. Although the authors explained the depilatory cream is to penetrate the GCV solution into the skin, the detailed demonstration about this or other function of depilatory cream in this MS is needed in the results.

Response 3: Thank you for the comments. As you indicated, rat skin without depilatory cream treatment in Figures 5 looks similar to that in Figure 4 with depilatory cream in non-magnified views. However, we can confirm that rat skin in Figures 5 remains very short hair shafts compared with that in Figure 4 in the magnified views of each photo.

Also, unfortunately we did not investigate the effects of depilatory cream on the permeability of GCV solution in details, therefore we have revised this description (Line 102).

Point 4: Without depilatory cream treatment, the GCV did not induced dermatitis (Figure 5), however the GCV also did not enter the skin. Thus, a WT mice group with depilatory cream and GCV treatment is crucial.

Response 4: Thank you for the comments. The reviewer has pointed that GCV cannot penetrate into the skin of Tg rats without treatment of depilatory cream. However, we showed that application of GCV without depilatory cream treatment induced elimination of NG2+ hair follicle stem cells and provoked contact dermatitis under application of DNFB in Figure 6. These data suggested that GCV penetrate into the skin of Tg rats without application of depilatory cream. Moreover, according to your comments, we confirmed that application of depilatory cream and GCV never showed the allergic responses in both wild type (WT) and normal Wistar rats. Also, we have described this results in “Results” section (Line 107-109).

Point 5: Figure 6, contact dermatitis marker gene expression analysis in two groups is helpful for the conclusion.

Response 5: Thank you for the suggestion. According to your comments, we performed qPCR studies to assess the gene expression of contact dermatitis markers in two groups: vehicle- and GCV-treated skin with treatment of DNFB or depilatory cream. We described this data in “Discussion” section (Line 184-187).

Round 2

Reviewer 1 Report

I thank the authors for addressing my concerns

Author Response

We deeply appreciated the referees’ comment.

Reviewer 2 Report

Generally, the authors answered most of my concerns except some results need to be shown in figures.

Line 88-92, the author claimed “The expression of NG2 in GCV-treated skin was less than half of that in vehicle-treated skin”. It is recommended to show this result in figures.

Line 184-187, the author also claimed the high CD4 and CD8 genes in GCV-treated skin. It is recommended to show this result in figures.

Author Response

Thank you for your comments and suggestions.

We would like to reply to the comments as below.

We have revised our manuscript in all parts indicated in red in the revised version.

Generally, the authors answered most of my concerns except some results need to be shown in figures.

Line 88-92, the author claimed “The expression of NG2 in GCV-treated skin was less than half of that in vehicle-treated skin”. It is recommended to show this result in figures.

Line 184-187, the author also claimed the high CD4 and CD8 genes in GCV-treated skin. It is recommended to show this result in figures.

Response: Thank you for the suggestion. According to your comments, we have added these results in Figure 3 and Figure 7, respectively. Also, we have added the description in “Figure legend” sections (Line 98-100 and Line 172-174).
